# Recurrent World Models Facilitate Policy Evolution

**David Ha**
Google Brain
Tokyo, Japan
hadavid@google.com

**Jürgen Schmidhuber**
NNAISENSE
The Swiss AI Lab, IDSIA (USI & SUPSI)
juergen@idsia.ch

## Abstract

A generative recurrent neural network is quickly trained in an unsupervised manner to model popular reinforcement learning environments through compressed spatio-temporal representations. The world model's extracted features are fed into compact and simple policies trained by evolution, achieving state of the art results in various environments. We also train our agent entirely inside of an environment generated by its own internal world model, and transfer this policy back into the actual environment. Interactive version of paper: `https://worldmodels.github.io`

## 1  Introduction

Humans develop a mental model of the world based on what they are able to perceive with their limited senses, learning abstract representations of both spatial and temporal aspects of sensory inputs. For instance, we are able to observe a scene and remember an abstract description thereof [7, 67]. Our decisions and actions are influenced by our internal predictive model. For example, what we perceive at any given moment seems to be governed by our predictions of the future [59, 52]. One way of understanding the predictive model inside our brains is that it might not simply be about predicting the future in general, but predicting future sensory data given our current motor actions [38, 48]. We are able to instinctively act on this predictive model and perform fast reflexive behaviours when we face danger [55], without the need to consciously plan out a course of action [52].

For many reinforcement learning (RL) problems [37, 96, 106], an artificial RL agent may also benefit from a predictive model (M) of the future [104, 95] (model-based RL). The backpropagation algorithm [50, 39, 103] can be used to train a large M in form of a neural network (NN). In partially observable environments, we can implement M through a recurrent neural network (RNN) [74, 75, 78, 49] to allow for better predictions based on memories of previous observation sequences.

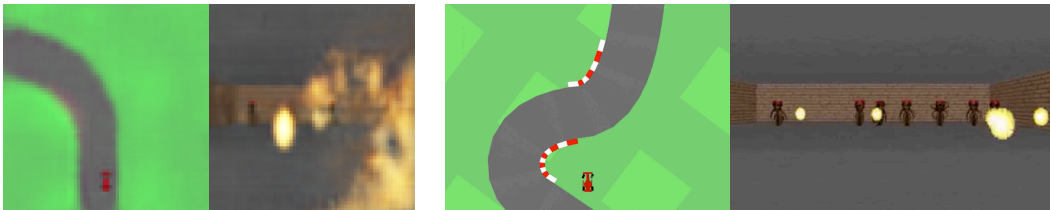

Figure 1: We build probabilistic generative models of OpenAI Gym [5] environments. These models can mimic the actual environments (left). We test trained policies in the actual environments (right).

In fact, our M will be a large RNN that learns to predict the future given the past in an unsupervised manner. M's internal representations of memories of past observations and actions are perceived and exploited by another NN called the controller (C) which learns through RL to perform some task without a teacher. A small and simple C limits C's credit assignment problem to a comparatively small search space, without sacrificing the capacity and expressiveness of the large and complex M.

We combine several key concepts from a series of papers from 1990–2015 on RNN-based world models and controllers [74, 75, 78, 76, 83] with more recent tools from probabilistic modelling, and present a simplified approach to test some of those key concepts in modern RL environments [5]. Experiments show that our approach can be used to solve a challenging race car navigation from pixels task that previously has not been solved using more traditional methods.

Most existing model-based RL approaches learn a model of the RL environment, but still train on the actual environment. Here, we also explore fully replacing an actual RL environment with a generated one, training our agent's controller C only inside of the environment generated by its own internal world model M, and transfer this policy back into the actual environment.

To overcome the problem of an agent exploiting imperfections of the generated environments, we adjust a *temperature* parameter of M to control the amount of uncertainty of the generated environments. We train C inside of a noisier and more uncertain version of its generated environment, and demonstrate that this approach helps prevent C from taking advantage of the imperfections of M. We will also discuss other related works in the model-based RL literature that share similar ideas of learning a dynamics model and training an agent using this model.

## 2   Agent Model

Our simple model is inspired by our own cognitive system. Our agent has a visual sensory component V that compresses what it sees into a small representative code. It also has a memory component M that makes predictions about future codes based on historical information. Finally, our agent has a decision-making component C that decides what actions to take based only on the representations created by its vision and memory components.

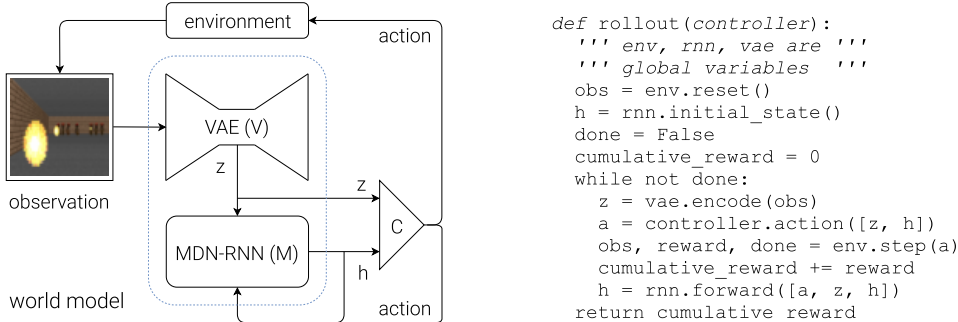

```
def rollout(controller):
    ''' env, rnn, vae are '''
    ''' global variables  '''
    obs = env.reset()
    h = rnn.initial_state()
    done = False
    cumulative_reward = 0
    while not done:
        z = vae.encode(obs)
        a = controller.action([z, h])
        obs, reward, done = env.step(a)
        cumulative_reward += reward
        h = rnn.forward([a, z, h])
    return cumulative_reward
```

Figure 2: Flow diagram showing how V, M, and C interacts with the environment (left). Pseudocode for how our agent model is used in the OpenAI Gym [5] environment (right).

Let the agent's life span be defined as a sequence of time steps, $t = 1, 2, \ldots, t_{\text{done}}$. Let $N_z$, $N_a$, $N_h$ be positive integer constants. The environment provides our agent with a high dimensional input observation at each time step $t$. This input is usually a 2D image frame that is part of a video sequence. The role of V is to learn an abstract, compressed representation of each observed input at each time step. Here, we use a Variational Autoencoder (VAE) [42, 71] as V to compress an image observed at time step $t$ into a latent vector $z_t \in \mathbb{R}^{N_z}$, with $N_z$ being a hyperparameter. While V's role is to compress what the agent sees at each time step, we also want to compress what happens over time. The RNN M serves as a predictive model of future $z_t$ vectors that V is expected to produce. Since many complex environments are stochastic in nature, we train our RNN to output a probability density function $p(z_t)$ instead of a deterministic prediction of $z_t$.

The agent takes an action $a_t \in \mathbb{R}^{N_a}$ at time $t$, where $N_a$ is the dimension of the action space. In our approach, we approximate $p(z_t)$ as a mixture of Gaussian distribution, and train M to output the probability distribution of the next latent vector $z_{t+1}$ given the current and past information made available to it. More specifically, the RNN, with $N_h$ hidden units, will model $P(z_{t+1} \mid a_t, z_t, h_t)$, where $h_t \in \mathbb{R}^{N_h}$ is the hidden state of the RNN at time step $t$. During sampling, we can adjust a real-valued *temperature* parameter $\tau$ to control model uncertainty, as done in previous work [28]. We will find that adjusting $\tau$ to be useful for training our controller later on. This approach is known as a Mixture Density Network [3] combined with an RNN (MDN-RNN) [24], and has been applied in the past for sequence generation problems such as generating handwriting [24, 6] and sketches [28].

C is responsible for determining the course of actions to take in order to maximize the expected cumulative reward of the agent during a rollout of the environment. In our experiments, we deliberately make C as simple and small as possible, and train it separately from V and M, so that most of our agent's complexity resides in V and M. C is a simple single layer linear model that maps $z_t$ and $h_t$ directly to action $a_t$ at each time step: $a_t = W_c\,[z_t\ h_t]\ + b_c$, where $W_c \in \mathbb{R}^{N_a \times (N_z + N_h)}$ and $b_c \in \mathbb{R}^{N_a}$ are the parameters that map the concatenated input $[z_t\ h_t]$ to the output action $a_t$.

This minimal design for C also offers important practical benefits. Advances in deep learning provided us with the tools to train large, sophisticated models efficiently, provided we can define a well-behaved, differentiable loss function. V and M are designed to be trained efficiently with the backpropagation algorithm using modern GPU accelerators, so we would like most of the model's complexity, and model parameters to reside in V and M. The number of parameters of C, a linear model, is minimal in comparison. This choice allows us to explore more unconventional ways to train C – for example, even using evolution strategies (ES) [70, 87] to tackle more challenging RL tasks where the credit assignment problem is difficult.

To optimize the parameters of C, we chose the Covariance-Matrix Adaptation Evolution Strategy (CMA-ES) [29, 30] as our optimization algorithm since it is known to work well for solution spaces of up to a few thousand parameters. We evolve parameters of C on a single machine with multiple CPU cores running multiple rollouts of the environment in parallel. For more information about the models, training procedures, and experiment configurations, please see the Supplementary Materials.

# 3 Car Racing Experiment: World Model for Feature Extraction

In this section, we describe how we can train the Agent model described earlier to solve a car racing task. To our knowledge, our agent is the first known to solve this task.[1]

Frame compressor V and predictive model M can help us extract useful representations of space and time. By using these features as inputs of C, we can train a compact C to perform a continuous control task, such as learning to drive from pixel inputs for a top-down car racing environment called `CarRacing-v0` [44]. In this environment, the tracks are randomly generated for each trial, and our agent is rewarded for visiting as many tiles as possible in the least amount of time. The agent controls three continuous actions: steering left/right, acceleration, and brake.

---

**Algorithm 1** Training procedure in our experiments.

    1. Collect 10,000 rollouts from a random policy.
    2. Train VAE (V) to encode frames into $z \in \mathbb{R}^{N_z}$.
    3. Train MDN-RNN (M) to model $P(z_{t+1} \mid a_t, z_t, h_t)$.
    4. Evolve controller (C) to maximize the expected cumulative reward of a rollout.

---

To train V, we first collect a dataset of 10k random rollouts of the environment. We have first an agent acting randomly to explore the environment multiple times, and record the random actions $a_t$ taken and the resulting observations from the environment. We use this dataset to train our VAE to encode each frame into low dimensional latent vector $z$ by minimizing the difference between a given frame and the reconstructed version of the frame produced by the decoder from $z$. We can now use our trained V to pre-process each frame at time $t$ into $z_t$ to train our M. Using this pre-processed data, along with the recorded random actions $a_t$ taken, our MDN-RNN can now be trained to model $P(z_{t+1} \mid a_t, z_t, h_t)$ as a mixture of Gaussians. [2]

In this experiment, V and M have no knowledge about the actual reward signals from the environment. Their task is simply to compress and predict the sequence of image frames observed. Only C has access to the reward information from the environment. Since there are a mere 867 parameters inside the linear C, evolutionary algorithms such as CMA-ES are well suited for this optimization task.

## 3.1 Experiment Results

*V without M*

Training an agent to drive is not a difficult task if we have a good representation of the observation. Previous works [35, 46] have shown that with a good set of hand-engineered information about the observation, such as LIDAR information, angles, positions and velocities, one can easily train a small feed-forward network to take this hand-engineered input and output a satisfactory navigation policy. For this reason, we first want to test our agent by handicapping C to only have access to V but not M, so we define our controller as $a_t = W_c\, z_t\, +\, b_c$.

Although the agent is still able to navigate the race track in this setting, we notice it wobbles around and misses the tracks on sharper corners, e.g., see Figure 1 (right). This handicapped agent achieved an average score of $632 \pm 251$, in line with the performance of other agents on OpenAI Gym's leaderboard [44] and traditional Deep RL methods such as A3C [41, 36]. Adding a hidden layer to C's policy network helps to improve the results to $788 \pm 141$, but not enough to solve this environment.

Table 1: `CarRacing-v0` results over 100 trials.

| Method | Average Score |
|---|---|
| DQN [66] | $343 \pm 18$ |
| A3C (continuous) [36] | $591 \pm 45$ |
| A3C (discrete) [41] | $652 \pm 10$ |
| Gym Leader [44] | $838 \pm 11$ |
| V model | $632 \pm 251$ |
| V model with hidden layer | $788 \pm 141$ |
| **Full World Model** | $\mathbf{906 \pm 21}$ |

Table 2: `DoomTakeCover-v0` results, varying $\tau$.

| Temperature $\tau$ | Virtual Score | Actual Score |
|---|---|---|
| 0.10 | $2086 \pm 140$ | $193 \pm 58$ |
| 0.50 | $2060 \pm 277$ | $196 \pm 50$ |
| 1.00 | $1145 \pm 690$ | $868 \pm 511$ |
| 1.15 | $918 \pm 546$ | $\mathbf{1092 \pm 556}$ |
| 1.30 | $732 \pm 269$ | $753 \pm 139$ |
| Random Policy | N/A | $210 \pm 108$ |
| Gym Leader [62] | N/A | $820 \pm 58$ |

*Full World Model (V and M)*

The representation $z_t$ provided by V only captures a representation at a moment in time and does not have much predictive power. In contrast, M is trained to do one thing, and to do it really well, which is to predict $z_{t+1}$. Since M's prediction of $z_{t+1}$ is produced from the RNN's hidden state $h_t$ at time $t$, $h_t$ is a good candidate for a feature vector we can give to our agent. Combining $z_t$ with $h_t$ gives C a good representation of both the current observation, and what to expect in the future.

We see that allowing the agent to access both $z_t$ and $h_t$ greatly improves its driving capability. The driving is more stable, and the agent is able to seemingly attack the sharp corners effectively. Furthermore, we see that in making these fast reflexive driving decisions during a car race, the agent does not need to *plan ahead* and roll out hypothetical scenarios of the future. Since $h_t$ contain information about the probability distribution of the future, the agent can just re-use the RNN's internal representation instinctively to guide its action decisions. Like a Formula One driver or a baseball player hitting a fastball [52], the agent can instinctively predict when and where to navigate in the heat of the moment.

Our agent is able to achieve a score of $906 \pm 21$, effectively solving the task and obtaining new state of the art results. Previous attempts [41, 36] using Deep RL methods obtained average scores of 591–652 range, and the best reported solution on the leaderboard obtained an average score of 838 $\pm$ 11. Traditional Deep RL methods often require pre-processing of each frame, such as employing edge-detection [36], in addition to stacking a few recent frames [41, 36] into the input. In contrast, our agent's V and M take in a stream of raw RGB pixel images and directly learn a spatio-temporal representation. To our knowledge, our method is the first reported solution to solve this task.

Since our agent's world model is able to model the future, we can use it to come up with hypothetical car racing scenarios on its own. We can use it to produce the probability distribution of $z_{t+1}$ given the current states, sample a $z_{t+1}$ and use this sample as the real observation. We can put our trained C back into this generated environment. Figure 1 (left) shows a screenshot of the generated car racing environment. The interactive version of this work includes a demo of the generated environments.

# 4 VizDoom Experiment: Learning Inside of a Generated Environment

We have just seen that a policy learned inside of the real environment appears to somewhat function inside of the generated environment. This begs the question – can we train our agent to learn inside of its own generated environment, and transfer this policy back to the actual environment?

If our world model is sufficiently accurate for its purpose, and complete enough for the problem at hand, we should be able to substitute the actual environment with this world model. After all, our agent does not directly observe the reality, but merely sees what the world model lets it see. In this experiment, we train an agent inside the environment generated by its world model trained to mimic a VizDoom [40] environment. In `DoomTakeCover-v0` [62], the agent must learn to avoid fireballs shot by monsters from the other side of the room with the sole intent of killing the agent. The cumulative reward is defined to be the number of time steps the agent manages to stay alive during a rollout. Each rollout of the environment runs for a maximum of 2100 time steps, and the task is considered solved if the average survival time over 100 consecutive rollouts is greater than 750 time steps.

## 4.1 Experiment Setup

The setup of our VizDoom experiment is largely the same as the Car Racing task, except for a few key differences. In the Car Racing task, M is only trained to model the next $z_t$. Since we want to build a world model we can train our agent in, our M model here will also predict whether the agent dies in the next frame (as a binary event $done_t$), in addition to the next frame $z_t$.

Since M can predict the $done$ state in addition to the next observation, we now have all of the ingredients needed to make a full RL environment to mimic `DoomTakeCover-v0` [62]. We first build an OpenAI Gym environment interface by wrapping a `gym.Env` [5] interface over our M as if it were a real Gym environment, and then train our agent inside of this *virtual* environment instead of using the actual environment. Thus in our simulation, we do not need the V model to encode any real pixel frames during the generation process, so our agent will therefore only train entirely in a more efficient latent space environment. Both virtual and actual environments share an identical interface, so after the agent learns a satisfactory policy inside of the virtual environment, we can easily deploy this policy back into the actual environment to see how well the policy transfers over.

Here, our RNN-based world model is trained to mimic a complete game environment designed by human programmers. By learning only from raw image data collected from random episodes, it learns how to simulate the essential aspects of the game, such as the game logic, enemy behaviour, physics, and also the 3D graphics rendering. We can even *play* inside of this generated environment.

Unlike the actual game environment, however, we note that it is possible to add extra uncertainty into the virtual environment, thus making the game more challenging in the generated environment. We can do this by increasing the temperature $\tau$ parameter during the sampling process of $z_{t+1}$. By increasing the uncertainty, our generated environment becomes more difficult compared to the actual environment. The fireballs may move more randomly in a less predictable path compared to the actual game. Sometimes the agent may even die due to sheer misfortune, without explanation.

After training, our controller learns to navigate around the virtual environment and escape from deadly fireballs launched by monsters generated by M. Our agent achieved an average score of 918 time steps in the virtual environment. We then took the agent trained inside of the virtual environment and tested its performance on the original VizDoom environment. The agent obtained an average score of 1092 time steps, far beyond the required score of 750 time steps, and also much higher than the score obtained inside the more difficult virtual environment. The full results are listed in Table 2.

We see that even though V is not able to capture all of the details of each frame correctly, for instance, getting the number of monsters correct, C is still able to learn to navigate in the real environment. As the virtual environment cannot even keep track of the exact number of monsters in the first place, an agent that is able to survive a noisier and uncertain generated environment can thrive in the original, cleaner environment. We also find agents that perform well in higher temperature settings generally perform better in the normal setting. In fact, increasing $\tau$ helps prevent our controller from taking advantage of the imperfections of our world model. We will discuss this in depth in the next section.

## 4.2 Cheating the World Model

In our childhood, we may have encountered ways to exploit video games in ways that were not intended by the original game designer [9]. Players discover ways to collect unlimited lives or health, and by taking advantage of these exploits, they can easily complete an otherwise difficult game. However, in the process of doing so, they may have forfeited the opportunity to learn the skill required to master the game as intended by the game designer. In our initial experiments, we noticed that our agent discovered an *adversarial* policy to move around in such a way so that the monsters in this virtual environment governed by M never shoots a single fireball during some rollouts. Even when there are signs of a fireball forming, the agent moves in a way to *extinguish* the fireballs.

Because M is only an approximate probabilistic model of the environment, it will occasionally generate trajectories that do not follow the laws governing the actual environment. As we previously pointed out, even the number of monsters on the other side of the room in the actual environment is not exactly reproduced by M. For this reason, our world model will be exploitable by C, even if such exploits do not exist in the actual environment.

As a result of using M to generate a virtual environment for our agent, we are also giving the controller access to all of the hidden states of M. This is essentially granting our agent access to all of the internal states and memory of the game engine, rather than only the game observations that the player gets to see. Therefore our agent can efficiently explore ways to directly manipulate the hidden states of the game engine in its quest to maximize its expected cumulative reward. The weakness of this approach of learning a policy inside of a learned dynamics model is that our agent can easily find an adversarial policy that can fool our dynamics model – it will find a policy that looks good under our dynamics model, but will fail in the actual environment, usually because it visits states where the model is wrong because they are away from the training distribution.

This weakness could be the reason that many previous works that learn dynamics models of RL environments do not actually use those models to fully replace the actual environments [60, 8]. Like in the M model proposed in [74, 75, 78], the dynamics model is deterministic, making it easily exploitable by the agent if it is not perfect. Using Bayesian models, as in PILCO [10], helps to address this issue with the uncertainty estimates to some extent, however, they do not fully solve the problem. Recent work [57] combines the model-based approach with traditional model-free RL training by first initializing the policy network with the learned policy, but must subsequently rely on model-free methods to fine-tune this policy in the actual environment.

To make it more difficult for our C to exploit deficiencies of M, we chose to use the MDN-RNN as the dynamics model of the *distribution* of possible outcomes in the actual environment, rather than merely predicting a deterministic future. Even if the actual environment is deterministic, the MDN-RNN would in effect approximate it as a stochastic environment. This has the advantage of allowing us to train C inside a more stochastic version of any environment – we can simply adjust the temperature parameter $\tau$ to control the amount of randomness in M, hence controlling the tradeoff between realism and exploitability.

Using a mixture of Gaussian model may seem excessive given that the latent space encoded with the VAE model is just a single diagonal Gaussian distribution. However, the discrete modes in a mixture density model are useful for environments with random discrete events, such as whether a monster decides to shoot a fireball or stay put. While a single diagonal Gaussian might be sufficient to encode individual frames, an RNN with a mixture density output layer makes it easier to model the logic behind a more complicated environment with discrete random states.

For instance, if we set the temperature parameter to a very low value of $\tau = 0.1$, effectively training our C with an M that is almost identical to a deterministic LSTM, the monsters inside this generated environment fail to shoot fireballs, no matter what the agent does, due to mode collapse. M is not able to transition to another mode in the mixture of Gaussian model where fireballs are formed and shot. Whatever policy learned inside of this generated environment will achieve a perfect score of 2100 most of the time, but will obviously fail when unleashed into the harsh reality of the actual world, underperforming even a random policy.

By making the temperature $\tau$ an adjustable parameter of M, we can see the effect of training C inside of virtual environments with different levels of uncertainty, and see how well they transfer over to the actual environment. We experiment with varying $\tau$ of the virtual environment, training an agent inside of this virtual environment, and observing its performance when inside the actual environment.

In Table 2, while we see that increasing $\tau$ of M makes it more difficult for C to find adversarial policies, increasing it too much will make the virtual environment too difficult for the agent to learn anything, hence in practice it is a hyperparameter we can tune. The temperature also affects the types of strategies the agent discovers. For example, although the best score obtained is $1092 \pm 556$ with $\tau = 1.15$, increasing $\tau$ a notch to 1.30 results in a lower score but at the same time a less risky strategy with a lower variance of returns. For comparison, the best reported score [62] is $820 \pm 58$.

## 5   Related Work

There is extensive literature on learning a dynamics model, and using this model to train a policy. Many basic concepts first explored in the 1980s for feed-forward neural networks (FNNs) [104, 56, 72, 105, 58] and in the 1990s for RNNs [74, 75, 78, 76] laid some of the groundwork for *Learning to Think* [83]. The more recent PILCO [10, 53] is a probabilistic model-based search policy method designed to solve difficult control problems. Using data collected from the environment, PILCO uses a Gaussian process (GP) model to learn the system dynamics, and uses this model to sample many trajectories in order to train a controller to perform a desired task, such as swinging up a pendulum.

While GPs work well with a small set of low dimension data, their computational complexity makes them difficult to scale up to model a large history of high dimensional observations. Other recent works [17, 12] use Bayesian neural networks instead of GPs to learn a dynamics model. These methods have demonstrated promising results on challenging control tasks [32], where the states well defined, and the observation is relatively low dimensional. Here we are interested in modelling dynamics observed from high dimensional visual data, as a sequence of raw pixel frames.

In robotic control applications, the ability to learn the dynamics of a system from observing only camera-based video inputs is a challenging but important problem. Early work on RL for active vision trained an FNN to take the current image frame of a video sequence to predict the next frame [85], and use this predictive model to train a fovea-shifting control network trying to find targets in a visual scene. To get around the difficulty of training a dynamical model to learn directly from high-dimensional pixel images, researchers explored using neural networks to first learn a compressed representation of the video frames. Recent work along these lines [99, 100] was able to train controllers using the bottleneck hidden layer of an autoencoder as low-dimensional feature vectors to control a pendulum from pixel inputs. Learning a model of the dynamics from a compressed latent space enable RL algorithms to be much more data-efficient [15, 101].

Video game environments are also popular in model-based RL research as a testbed for new ideas. Previous work [51] used a feed-forward convolutional neural network (CNN) to learn a forward simulation model of a video game. Learning to predict how different actions affect future states in the environment is useful for game-play agents, since if our agent can predict what happens in the future given its current state and action, it can simply select the best action that suits its goal. This has been demonstrated not only in early work [58, 85] (when compute was a million times more expensive than today) but also in recent studies [13] on several competitive VizDoom environments.

The works mentioned above use FNNs to predict the next video frame. We may want to use models that can capture longer term time dependencies. RNNs are powerful models suitable for sequence modelling [24]. Using RNNs to develop internal models to reason about the future has been explored as early as 1990 [74], and then further explored in [75, 78, 76]. A more recent work [83] presented a unifying framework for building an RNN-based general problem solver that can learn a world model of its environment and also learn to reason about the future using this model. Subsequent works have used RNN-based models to generate many frames into the future [8, 60, 11, 25], and also as an internal model to reason about the future [90, 68, 102].

In this work, we used evolution strategies (ES) to train our controller, as this offers many benefits. For instance, we only need to provide the optimizer with the final cumulative reward, rather than the entire history. ES is also easy to parallelize – we can launch many instances of `rollout` with different solutions to many workers and quickly compute a set of cumulative rewards in parallel. Recent works [14, 73, 26, 94] have demonstrated that ES is a viable alternative to traditional Deep RL methods on many strong baselines. Before the popularity of Deep RL methods [54], evolution-based algorithms have been shown to be effective at solving RL tasks [92, 22, 21, 18, 88]. Evolution-based algorithms have even been able to solve difficult RL tasks from high dimensional pixel inputs [45, 31, 63, 1].

# 6 Discussion

We have demonstrated the possibility of training an agent to perform tasks entirely inside of its simulated latent space world. This approach offers many practical benefits. For instance, video game engines typically require heavy compute resources for rendering the game states into image frames, or calculating physics not immediately relevant to the game. We may not want to waste cycles training an agent in the actual environment, but instead train the agent as many times as we want inside its simulated environment. Agents that are trained incrementally to simulate reality may prove to be useful for transferring policies back to the real world. Our approach may complement *sim2real* approaches outlined in previous work [4, 33].

The choice of implementing V as a VAE and training it as a standalone model also has its limitations, since it may encode parts of the observations that are not relevant to a task. After all, unsupervised learning cannot, by definition, know what will be useful for the task at hand. For instance, our VAE reproduced unimportant detailed brick tile patterns on the side walls in the Doom environment, but failed to reproduce task-relevant tiles on the road in the Car Racing environment. By training together with an M that predicts rewards, the VAE may learn to focus on task-relevant areas of the image, but the tradeoff here is that we may not be able to reuse the VAE effectively for new tasks without retraining. Learning task-relevant features has connections to neuroscience as well. Primary sensory neurons are released from inhibition when rewards are received, which suggests that they generally learn task-relevant features, rather than just any features, at least in adulthood [65].

In our experiments, the tasks are relatively simple, so a reasonable world model can be trained using a dataset collected from a random policy. But what if our environments become more sophisticated? In any difficult environment, only parts of the world are made available to the agent only after it learns how to strategically navigate through its world. For more complicated tasks, an iterative training procedure is required. We need our agent to be able to explore its world, and constantly collect new observations so that its world model can be improved and refined over time. Future work will incorporate an iterative training procedure [83], where our controller actively explores parts of the environment that is beneficial to improve its world model. An exciting research direction is to look at ways to incorporate artificial curiosity and intrinsic motivation [81, 80, 77, 64, 61] and information seeking [86, 23] abilities in an agent to encourage exploration [47]. In particular, we can augment the reward function based on improvement in compression quality [81, 80, 77, 83].

Another concern is the limited capacity of our world model. While modern storage devices can store large amounts of historical data generated using an iterative training procedure, our LSTM [34, 20]-based world model may not be able to store all of the recorded information inside of its weight connections. While the human brain can hold decades and even centuries of memories to some resolution [2], our neural networks trained with backpropagation have more limited capacity and suffer from issues such as catastrophic forgetting [69, 16, 43]. Future work will explore replacing the VAE and MDN-RNN with higher capacity models [89, 27, 93, 97, 98], or incorporating an external memory module [19, 107], if we want our agent to learn to explore more complicated worlds.

Like early RNN-based C–M systems [74, 75, 78, 76], ours simulates possible futures time step by time step, without profiting from human-like hierarchical planning or abstract reasoning, which often ignores irrelevant spatio-temporal details. However, the more general *Learning To Think* [83] approach is not limited to this rather naive approach. Instead it allows a recurrent C to learn to address *subroutines* of the recurrent M, and reuse them for problem solving in arbitrary computable ways, e.g., through hierarchical planning or other kinds of exploiting parts of M's program-like weight matrix. A recent *One Big Net* [84] extension of the C–M approach collapses C and M into a single network, and uses PowerPlay-like [82, 91] behavioural replay (where the behaviour of a teacher net is compressed into a student net [79]) to avoid forgetting old prediction and control skills when learning new ones. Experiments with those more general approaches are left for future work.

**Acknowledgments**

We would like to thank Blake Richards, Kory Mathewson, Chris Olah, Kai Arulkumaran, Denny Britz, Kyle McDonald, Ankur Handa, Elwin Ha, Nikhil Thorat, Daniel Smilkov, Alex Graves, Douglas Eck, Mike Schuster, Rajat Monga, Vincent Vanhoucke, Jeff Dean and Natasha Jaques for their thoughtful feedback. This work was partially funded by SNF project RNNAISSANCE (200021_165675) and by an ERC Advanced Grant (no: 742870).

## Footnotes

[1] We find this task interesting because although it is not difficult to train an agent to wobble around randomly generated tracks and obtain a mediocre score, `CarRacing-v0` defines *solving* as getting average reward of 900 over 100 consecutive trials, which means the agent can only afford very few driving mistakes.

[2] Although in principle, we can train V and M together in an end-to-end manner, we found that training each separately is more practical, achieves satisfactory results, and does not require exhaustive hyperparameter tuning. As images are not required to train M on its own, we can even train on large batches of long sequences of latent vectors encoding the entire 1000 frames of an episode to capture longer term dependencies, on a single GPU.

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
