[Supplementary Material]

# A  Supplementary Materials

In this section we will describe in more details the models and training methods used in this work.

## A.1  Comparing V, M, C Model Sizes

Table 1: `CarRacing-v0` Parameter Count

| MODEL | PARAMETER COUNT |
|---|---|
| VAE | 4,348,547 |
| MDN-RNN | 422,368 |
| CONTROLLER | 867 |

Table 2: `DoomTakeCover-v0` Parameter Count

| MODEL | PARAMETER COUNT |
|---|---|
| VAE | 4,446,915 |
| MDN-RNN | 1,678,785 |
| CONTROLLER | 1,088 |

## A.2  Variational Autoencoder

Figure 1: Description of tensor shapes for each layer of our ConvVAE. (left).
MDN-RNN similar to the one used in [2, 3, 5] (right).

We trained a Convolutional Variational Autoencoder (ConvVAE) model as our agent's V, as illustrated in Figure 1 (left). Unlike vanilla autoencoders, enforcing a Gaussian prior over the latent vector $z_t$ also limits the amount of information capacity for compressing each frame, but this Gaussian prior also makes the world model more robust to unrealistic $z_t \in \mathbb{R}^{N_z}$ vectors generated by M.

Our latent vector $z_t$ is sampled from a factored Gaussian distribution $N(\mu_t, \sigma_t^2 I)$, with mean $\mu_t \in \mathbb{R}^{N_z}$ and diagonal variance $\sigma_t^2 \in \mathbb{R}^{N_z}$. As the environment may give us observations as high dimensional pixel images, we first resize each image to 64x64 pixels and use this resized image as V's observation. Each pixel is stored as three floating point values between 0 and 1 to represent each of the RGB channels. The ConvVAE takes in this 64x64x3 input tensor and passes it through 4 convolutional layers to encode it into low dimension vectors $\mu_t$ and $\sigma_t$. In the Car Racing task, $N_z$ is 32 while for the Doom task $N_z$ is 64. The latent vector $z_t$ is passed through 4 of deconvolution layers used to decode and reconstruct the image.

Each convolution and deconvolution layer uses a stride of 2. The layers are indicated in the diagram in *Italics* as *Activation-type Output Channels x Filter Size*. All convolutional and deconvolutional layers use relu activations except for the output layer as we need the output to be between 0 and 1. We trained the model for 1 epoch over the data collected from a random policy, using $L^2$ distance between the input image and the reconstruction to quantify the reconstruction loss we optimize for, in addition to KL loss.

### A.3 Mixture Density Network + Recurrent Neural Network

To implement M, we use an LSTM [7] recurrent neural network combined with a Mixture Density Network [1] as the output layer, as illustrated in Figure 1 (right). We use this network to model the probability distribution of $z_t$ as a Mixture of Gaussian distribution. This approach is very similar to previous work [3] in the Unconditional Handwriting Generation section and also the decoder-only section of SketchRNN [5]. The only difference is that we did not model the correlation parameter between each element of $z_t$, and instead had the MDN-RNN output a diagonal covariance matrix of a factored Gaussian distribution.

Unlike the handwriting and sketch generation works, rather than using the MDN-RNN to model the probability density function (pdf) of the next pen stroke, we model instead the pdf of the next latent vector $z_t$. We would sample from this pdf at each time step to generate the environments. In the Doom task, we also use the MDN-RNN to predict the probability of whether the agent has died in this frame. If that probability is above 50%, then we set *done* to be *true* in the virtual environment. Given that death is a low probability event at each time step, we find the cutoff approach to be more stable compared to sampling from the Bernoulli distribution.

The MDN-RNNs were trained for 20 epochs on the data collected from a random policy agent. In the Car Racing task, the LSTM used 256 hidden units, in the Doom task 512 hidden units. In both tasks, we used 5 Gaussian mixtures, but unlike [3, 5], we did not model the correlation parameters, hence $z_t$ is sampled from a factored mixture of Gaussian distributions.

When training the MDN-RNN using teacher forcing from the recorded data, we store a pre-computed set of $\mu_t$ and $\sigma_t$ for each of the frames, and sample an input $z_t \sim N(\mu_t, \sigma_t^2 I)$ each time we construct a training batch, to prevent overfitting our MDN-RNN to a specific sampled $z_t$.

### A.4 Controller

For both environments, we applied $\tanh$ nonlinearities to clip and bound the action space to the appropriate ranges. For instance, in the Car Racing task, the steering wheel has a range from -1.0 to 1.0, the acceleration pedal from 0.0 to 1.0, and the brakes from 0.0 to 1.0. In the Doom environment, we converted the discrete actions into a continuous action space between -1.0 to 1.0, and divided this range into thirds to indicate whether the agent is moving left, staying where it is, or moving to the right. We would give C a feature vector as its input, consisting of $z_t$ and the hidden state of the MDN-RNN. In the Car Racing task, this hidden state is the output vector $h_t \in \mathbb{R}^{N_h}$ of the LSTM, while for the Doom task it is both the cell vector $c_t \in \mathbb{R}^{N_h}$ and the output vector $h_t$ of the LSTM.

### A.5 Evolution Strategies

We used *Covariance-Matrix Adaptation Evolution Strategy* (CMA-ES) [6] to evolve C's weights. Following the approach described in *Evolving Stable Strategies* [4], we used a population size of 64, and had each agent perform the task 16 times with different initial random seeds. The agent's fitness value is the *average cumulative reward* of the 16 random rollouts. The diagram below (left) charts the best performer, worst performer, and mean fitness of the population of 64 agents at each generation:

Figure 2: Training progress of `CarRacing-v0` (left).
Histogram of cumulative rewards. Score is $906 \pm 21$ (right).

Since the requirement of this environment is to have an agent achieve an average score above 900 over 100 random rollouts, we took the best performing agent at the end of every 25 generations, and tested it over 1024 random rollout scenarios to record this average on the red line. After 1800 generations, an agent was able to achieve an average score of 900.46 over 1024 random rollouts. We used 1024 random rollouts rather than 100 because each process of the 64 core machine had been configured to run 16 times already, effectively using a full generation of compute after every 25 generations to evaluate the best agent 1024 times. In the Figure 3 (left) below, we plot the results of same agent evaluated over 100 rollouts:

Figure 3: When agent sees only $z_t$ but not $h_t$, score is $632 \pm 251$ (left).
If we add a hidden layer on top of only $z_t$, score increases to $788 \pm 141$ (right).

We also experimented with an agent that has access to only the $z_t$ vector from the VAE, but not the RNN's hidden states. We tried 2 variations, where in the first variation, C maps $z_t$ directly to the action space $a_t$. In second variation, we attempted to add a hidden layer with 40 $tanh$ activations between $z_t$ and $a_t$, increasing the number of model parameters of C to 1443, making it more comparable with the original setup. These results are shown in In the Figure 3 (right).

### A.6   DoomRNN

We conducted a similar experiment on the generated Doom environment we called *DoomRNN*. Please note that we did not attempt to train our agent on the actual VizDoom environment, but only used VizDoom for the purpose of collecting training data using a random policy. *DoomRNN* is more computationally efficient compared to VizDoom as it only operates in latent space without the need to render an image at each time step, and we do not need to run the actual Doom game engine.

Figure 4: Training of DoomRNN (left). Histogram of time steps survived in the actual VizDoom environment over 100 consecutive trials. Score is $1092 \pm 556$ (right).

In our virtual DoomRNN environment we increased the temperature slightly and used $\tau = 1.15$ to make the agent learn in a more challenging environment. The best agent managed to obtain an average score of 959 over 1024 random rollouts. This is the highest score of the red line in Figure 4 (left). This same agent achieved an average score of $1092 \pm 556$ over 100 random rollouts when deployed to the actual `DoomTakeCover-v0` [8] environment, as shown in Figure 4 (right).