[Reviews · NeurIPS 2018]

Reviewer 1



Summary: This paper proposes a new way to develop a world model for reinforcement learning. The focus is on the encoding of the visual world, coupled with a world model that learns based on the compressed representation. The world model is a recurrent version of Bishop’s (1995, neural networks book, chapter 6) mixture of gaussians network. That network outputs the weights of an MOG (using softmax), the means of the gaussians (linear outputs), and the variance (modeled as e^var, so it is a scale parameter). I had not seen a recurrent version of this network before. The model is very simple: 0) The world is explored using a random policy with many rollouts. 1) The visual world is encoded using a VAE using the visited images gotten in the rollout, call it V. 2) The world model is trained using a recurrent NN with a five-component MOG to predict the next hidden state of V given the current hidden state. Call it M. 3) A very simple (one or two layer) controller is trained using the current hidden state of V and the current hidden state of M, using an evolution strategie method (the Covariance-Matrix Adaptation Evolution Strategy (CMA-ES), which has been shown to work well in high dimensional spaces. The network achieves state of the art (“solves”) the OPENAI Gym CarRacingv0 task, the first system to do so. A slightly modified network also achieves state of the art on the DoomTakeCover-v0 task, by first training V using a random policy, and then training M, but then training the controller on it’s own world model, rather than training directly on the task. I.e., the world model is iterated to produce the next hidden state of V, without ever using V, and the controller learns from that. In order for this to work, the controller also has to return a prediction of the agent’s death, as that is how rewards are calculated. This is very cool - it is like a tennis player who simulates playing in his or her mind, and improves as a result. The other difference from the previous model is that the controller not only gets h from the world model, but the c’s as well (it’s an LSTM network). Quality: Strengths This is a well-written and well-reasoned paper, and the results are clear. The use of a variety of techniques from different authors shows a mastery of the field, and many approaches are discussed. As a result, the references are humongous for a NIPS paper: 106 references! This alone, while a bit over the top, creates a valuable reading list for new PhD students interested in RL. While many of the details of the networks and training are in the supplementary material (there is also an interactive web site I did not look at), I think this is ok. Weaknesses In the references, about 20% of them are to the author’s own work (it’s obvious who’s work this is). Dial back that ego, guy! ;-) Clarity: The paper is quite clear. Originality: The combination of techniques is novel. The work combines techniques from many researchers. Significance: This is in some ways, a very simple model that solves a difficult task. I plan to teach it in my next iteration of my deep learning class.

Reviewer 2



The authors present a solid and exciting continuation of decades long work into reinforcement learning using world modeling. The authors use a VAE to learn an efficient compression of the visual space (for racing and DOOM) without any exposure to reward signals. An RNN is then trained to predict the probabilistic distribution of the next compressed visual frame, while a small controller is trained to use both of these other networks to determine actions in the environment. Interestingly, when the world model is forced to tackle with uncertainty, the controller is able to learn a robust policy without any interaction with the true environment. The interactive demo website provided with the submission is one of the most unique, enjoyable, and informative presentations of a machine learning result I have ever encountered. All papers should be presented in this way! Major Comments: Have you compared CMA-ES to more standard ES with many more parameters? Do they yield similar performance? An MDN-RNN is used to increase the apparent stochasticity of the environment, which can be further manipulated through the temperature parameter, to avoid cheating. Do you think this construction is necessary, or could similar performance be achieved by simply adding gaussian noise to a deterministic RNN? I’m curious whether it’s necessary to train the RNN at all. Would a cleverly initialized RNN, i.e. to maintain a history of its inputs at a useful timescale, provide a sufficient richness of history for the controller?

Reviewer 3



Building the "word model" that predicts the future and helps take actions is very important and has been discussed for years in reinforcement learning as model-based RL. This paper combines several key concepts from a series of papers from 1990–2015 on RNN-based world models and controllers, with more recent tools from probabilistic modeling, and present a simplified approach to test some of those key concepts in modern RL environments. This paper also explores fully replacing an actual RL environment with a generated one, training the agent’s controller C only inside of the environment generated by its own internal world model M, and transfer this policy back into the actual environment. Question to novelty: using VAE to extract features, and using RNN or memory to form a policy are both proposed before in previous works. Details: -- Training the variational autoencoder (V), the RNN model (M) and the controller (C) separately makes the analysis easier. It also enables the model to learn policy only based its own prediction from M. What if these modules are optimized jointly? -- In algorithm1, the rollouts are initially collected randomly. Will this be difficult to apply to complicated environments where random policy will always fail (so that training M will be difficult)